# Identification of Important Wetlands and Optimization of Landscape Patterns Based on Human Pressure Index: The Case of the Linghekou Wetland in China

**Meiqing Wang, Qian Cheng * and Ruixin Chen**

College of Water Resource, Shenyang Agricultural University, Shenyang 110866, China;
wangmq3366@163.com (M.W.); baiyangchen5454@163.com (R.C.)
* Correspondence: chengqian1979@163.com

**Abstract:** The Linghekou wetland is a rich repository of ecological resources and serves as an important habitat for numerous rare and protected animals. However, due to a confluence of natural and anthropogenic factors, the ecological environment of the Linghekou wetland is facing a multitude of threats, including the reduction in wetland area, the degradation of wetland resources, and the instability of ecological structure. This paper employs an anthropogenic focus, utilizing the human pressure index (HPI), spatial autocorrelation, and cold and hot spot methods to identify crucial wetlands. These identified wetlands are then utilized as ecological source sites to optimize the landscape pattern of the Linghekou wetland, employing the minimum cumulative resistance (*MCR*) model. The final results indicated the identification of 6 ecological sources, 8 ecological corridors, and 42 ecological nodes. These were primarily concentrated in the southern region of the study area and were distributed in a reasonable manner. The method of identifying ecological sources when optimizing the landscape pattern with the *MCR* model was enriched by this approach. Additionally, the paper offers recommendations for the optimization of the landscape pattern of the Linghekou wetland and establishes a foundation for the protection and restoration of other similar wetlands.

**Keywords:** human pressure index; wetland; spatial autocorrelation; cold and hot spots; *MCR* model





## 1. Introduction

Wetlands represent a vital ecological resource in nature, offering a distinctive array of ecological services [1–4]. Estuarine wetlands represent a significant type of wetland ecosystem, playing a vital role in the maintenance of ecological balance. They serve as vital water retention and purification systems, provide flood storage and drought prevention capabilities, and act as climate regulators. Additionally, they serve as vital habitats for diverse species of flora and fauna [5,6]. Nevertheless, the accelerated expansion of coastal regions and the concomitant increase in population over the past 50 years have resulted in a precipitous decline in the extent of wetlands across the globe [7,8]. The two main drivers of the decline of natural wetlands are human activities and climate change. However, with the gradual development of the economy, human activities are intensifying, placing increasing pressure on wetlands. This pressure is affecting and altering the ecological structure and function of wetlands, potentially triggering ecological imbalance and environmental degradation. As a result, human activities have become the primary driver of wetland resource degradation [9–11]. Consequently, research is required to elucidate the relationship between human pressure and wetland resources, with the objective of maintaining the sustainability of wetland resources. The analysis of the impact of human pressure indices on the spatial distribution pattern of wetlands is of paramount importance for the conservation of wetlands and the sustainable development of the ecological environment.

In previous studies, two main methods have been employed to quantify human activity: Field survey and data collection [12–15]. Field survey methods include the an-

thropogenic activity index (AAI), the human disturbance activity index (HDAI), and the anthropogenic stress index (ASI), among others. The AAI is based on field surveys, with scores assigned by staff on the spot, which is more realistic. However, this approach has the disadvantages of inconsistent scoring standards and strong subjectivity [16]. The combination of the human disturbance activity index (HDAI) and the anthropogenic stress index (ASI) provides a more comprehensive assessment of human activities. However, the method relies on field observations and requires different observation stations depending on the landscape type. To ensure the accuracy of the results, all work must be completed in one day, which increases the consumption of manpower and materials and is not suitable for the study of the study area on a large scale [17]. The primary objective of data collection is to obtain information on the study area and quantify human activities within it. This is achieved through the processing of information and data [18,19]. The most prominent research methods employed are the anthropogenic pressure index on biomes (APIB), the Hemeroby index (HI), the area of influence of diffuse anthropization (AIDA), and the human pressure index (HPI). The APIB primarily relies on land-use data, and has experimentally demonstrated that land-use changes can be used to characterize anthropogenic disturbances and develop effective programs for the conservation of species communities [20]. The HI methodology considers anthropogenic influences primarily on landscape patterns, thus assigning scores based solely on landscape type. However, the scoring criteria have been enhanced to incorporate the spatial distribution of patches. Nevertheless, a more comprehensive characterization of anthropogenic activities is warranted [21]. The AIDA represents an innovative indicator of human impact. The methodology considers six different criteria, and experiments were conducted in three case areas. These demonstrated a correlation between AIDA and the degradation of protected areas. However, the study has limitations due to its initial stage. It has not yet been able to establish a categorization criterion that would be applicable to the majority of the study areas [22]. The HPI is a comprehensive evaluation index in landscape ecology. It is a quantitative indicator of the direct and indirect impacts of human activities on the ecological environment and is therefore an important indicator for assessing the environmental pressure caused by human activities [23–27]. The HPI does not require the implementation of field surveys, and the selection of indicators is more comprehensive when considering human activities; as evidenced by previous research, it is an effective tool for assessing the impact of human life and production activities on wetlands, offering a more comprehensive understanding than previous studies [28,29]. By incorporating the HPI into the study of wetland resource distribution, researchers can gain a more nuanced understanding of the impact of human activities on wetlands, which is crucial for the sustainable development of these ecosystems. The optimization of landscape patterns is currently being investigated through the use of computer optimization algorithms, circuit models, and minimum cumulative resistance (*MCR*) models. The computer optimization algorithm can combine the given conditions to propose the optimal optimization strategy. The model used can also be well fitted to the study area. However, in computer optimization, there is a lack of human consideration for the spatial and temporal characteristics of different plaques. Furthermore, there may be a divergence in the results of the optimal solution, which may lead to errors [30]. The circuit model treats species migration as completely random in the analysis, which aligns with the natural characteristics of species migration. This approach effectively identifies ecological landscapes and ecological pinch points, which play an important role in connectivity. However, it ignores the fact that species migration is not completely random. Some species will adapt to the local environment and choose migration paths that are favorable to them [31,32]. The *MCR* model can generate species migration routes with the least resistance. The *MCR* models have been employed in a variety of settings, including hilly regions [33], urban areas [34], wetlands [35], and rural communities [36]. The construction of ecological corridors based on the *MCR* model can effectively control human activities and reduce resistance to species migration, thereby achieving the goal of protecting species diversity [37–39].

The Linghekou wetland is a vital nature reserve within the estuary, comprising a plethora of ecological resources and serving a pivotal role in the conservation of protected and endangered animals [40]. In recent years, the accelerated development of wetlands has been observed due to changes in natural factors and human activities. This has caused a number of important problems, including a reduction in wetland areas, degradation of wetland resources, and instability of the ecological structure of existing wetlands [41]. Long-term assessment of ecosystem quality is essential for evaluating the stability of regional ecosystem services, maintaining ecological security, and promoting sustainable development. This study aims to investigate the correlation between human pressure and the distribution of wetlands in the Linghekou wetland by using the HPI. The study considers natural, economic, and social factors and combines spatial autocorrelation and cold and hot spot analysis methods. Areas with a high proportion of wetlands, distributed in clusters and occupying a large region, alongside those areas where human pressure persistently increases from year to year, suggest that wetlands play a crucial role in the evolutionary process. Such areas are considered significant wetlands in this paper. Based on the important wetland, the ecological source is determined, and the *MCR* model is implemented to develop the ecological corridor and optimize the landscape pattern. Furthermore, it is evident that the surrounding areas of wetlands require protection, with the objective of examining the laws of wetlands' evolution in response to human activities. This exploration aims to provide a scientific foundation for the conservation and utilization of wetlands, to comprehensively understand the impact of human activities on wetlands, and to suggest recommendations for the reasonable regulation of human activities.

## 2. Data and Method

### 2.1. Study Area

The study area is situated within the Linghekou wetland, which is part of the north coast of Liaodong Bay, Bohai Sea, Liaoning Province, China. It encompasses 68.7 km of the coastline of Linghai City, with geographical coordinates of 40°45′–41°00′ N and 121°00′–121°30′ E. The Linghekou wetland is an important part of the Liaohe Delta wetlands, covering an area of approximately 838.66 km$^2$. It is a mixed coastal wetland ecosystem. The wetland is influenced by marine factors such as tides, currents, and waves in the northern part of Liaodong Bay, forming large areas of tidal flats and marsh wetlands. This is highly conducive to wetland ecology. The reserve is home to a rich diversity of species and resources, with a total of 239 families and 1024 species of plants and animals. The Linghekou wetland is situated at the center of the Northeast Asian bird migration route, with tens of thousands of birds passing through annually. It is also home to numerous endangered species, such as cranes, which require a specific breeding environment and habitat for their survival. This makes the area an ideal location for wildlife conservation research. Figure 1 depicts the geographical location of the Linghekou wetland.

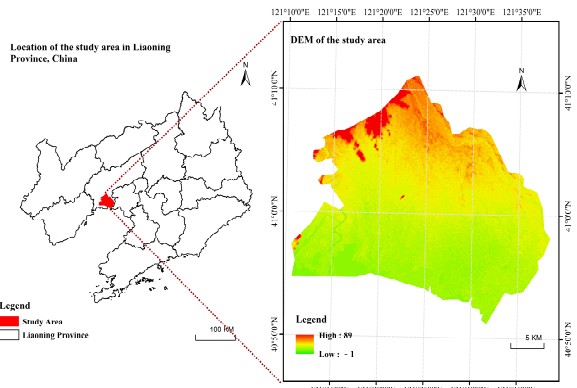

**Figure 1.** Geographical location of the Linghekou wetland.

### 2.2. Data Sources

The distribution of wetlands in the Linghekou wetland was obtained by generating a grid and calculating the proportion of wetland area within the grid (500 m × 500 m).

The HPI is based on the human footprint calculation methodology combined with actual local human impacts [42,43]. The components of the HPI include built-up areas, different types of land use, roads, railways, navigable water bodies, population density, and lighting levels [9,10,44,45]. The assignment rules are shown in Table 1.

**Table 1.** The HPI assignment classification.

| Assignment Range | Assignment Basis | Assignment | Assignment Calculation | Full Marks |
|---|---|---|---|---|
| Assigning values in terms of the area covered | (1) Built-up area | 10 | | 10 |
| | (2) Land-use classification | | | |
| | Paddy field | 8 | | |
| | Dry land | 7 | | 8 |
| | Farmed lakes | 7 | | |
| | Other | 0 | | |
| Assignment of values in terms of area covered and buffer range | (3) Roads | | | |
| | Within 0.5 km buffer zone | 8 | | |
| | Outside 0.5 km buffer zone | 4 | | 8 |
| | Within 0.5–15 km buffer zone | 0~4 | Decreasing exponentially from near to far | |
| | Outside the 15 km buffer zone | 0 | | |
| | (4) Railway | | | |
| | Within 0.5 km buffer zone | 8 | | 8 |
| | Outside 0.5 km buffer zone | 0 | | |
| | (5) Navigable water | | | |
| | Within a 15 km buffer zone of the coastline | 0~4 | Decreasing exponentially from near to far | 4 |
| | Within a 15 km buffer zone of a major river | 0~4 | Decreasing exponentially from near to far | |
| | Other | 0 | | |
| Assignment by density | (6) Population density | 10 | | |
| | >1000 people/km$^2$ | 10 | | 20 |
| | ≤1000 people/km$^2$ | 0~10 | Logarithmic adjustment of pressure scores | |
| | (7) Light value | 0~10 | Equivalent quintile method | |
| Full HPI score | | | | 58 |

(1) The landscape pattern data for the study area was obtained through image interpretation. Remote sensing image data was obtained through the geospatial data cloud and then subjected to supervised classification combined with manual visual interpretation. The interpretation accuracy reached over 90%, and the Kappa coefficient was 0.85. The built environment of the study area comprises residential areas, commercial buildings, and urban parks. However, due to its high level of human activity, it is not a suitable habitat for protected species and does not provide high ecosystem service value. Therefore, for the purposes of this paper, such areas are classified as built-up areas and assigned a pressure value of 10 [46].

(2) The manner in which land is used can have an impact on the ecosystem and the quality of habitats for different species. The extent to which land use affects ecosystem processes and habitat quality varies depending on the specific land use in question [9,47–49]. In this paper, the study area mainly contains several types of land use such as paddy fields, dry land, farmed lake, beach, reed swamp, and woodland, among which beach, woodland, and reed swamp include both natural and human modification and protection, but generally do not involve too much human behavior of building settlements, growing

food, and producing other economic products through land modification, i.e., they are assigned a value of 0. The three land-use types of paddy field, dry land, and farmed lake are assigned a value of 7 or 8, based on their level of human input, environmental pressure [10], and the extent and impact of altering ecosystem processes.

(3) The data utilized to assign values to the area covered and the buffer zone are derived from the Open Street Map. The distinction between the impact of motorways and unclassified roads on the surrounding ecological environment is not readily apparent, so no classification is assigned to them. The score is reduced exponentially from 0 to 4 for distances ranging from 0.5 km to 15 km, and 0 for distances exceeding 15 km [50,51].

(4) The 0.5 km buffer zone within the railway in the study area and outside the 0.5 km buffer zone was assigned a direct score, which ranged from 8 points for a maximum impact to 0 points for no impact.

(5) The data on navigable water use were obtained from the National Basic Geographic Database. The 68.7 km of coastline in Linghai City included in the geographical location of the study area decreases exponentially from 0 km from the navigable water body to an infinite distance of 15 km away until it is reduced to a score of 0. This is the distance from the navigable water body to the study area.

(6) The data utilized in this study were obtained from the National Population Project (WorldPop) and are presented at a resolution of 1000 m. In instances where the population density is greater than 1000 people/km$^2$, a score of 10 is assigned. Conversely, in instances where the population density is less than or equal to 1000 people/km$^2$, the pressure score is adjusted using the logarithmic method and assigned a score between 0 and 10.

Logarithmic method: pressure score = 3.33 × log (population density + 1)

(7) Night light data are corrected DMSP-OLS-like data obtained by integrating DMSP-OLS and SNPP-VIIRS data with a resolution of about 1000 m [52], scored from 0 to 10 using an equal five-digit method.

The value is based on the definition of the human footprint calculation method for each assignable item, combined with the actual situation of the study area, as reflected in its calculation method, and finally summed to obtain the HPI value. The value increases in proportion to the level of human pressure, with the highest score of 58 points corresponding to high human pressure and the lowest score of 0 points corresponding to no human pressure.

### 2.3. Spatial Autocorrelation Analysis

Based on ArcGIS vector data, spatial autocorrelation reflects the distribution of data for a certain attribute value on a spatial unit, based on the existence of correlation in the geospatial distribution. The fishnet tool was used to create a class of elements containing a network of rectangular image elements, and spatial autocorrelation and cold and hot spots were used for analysis.

The spatial autocorrelation (global Moran's *I*) tool measures spatial autocorrelation based on both element location and element value and is expressed as follows [29,53]:

$$I = \frac{n}{\sum\limits_{i=1}^{n} (x_i - \overline{x})^2} \times \frac{\sum\limits_{i=1}^{n} \sum\limits_{j=1}^{n} w_{ij}(x_i - \overline{x})(x_j - \overline{x})}{\sum\limits_{i=1}^{n} \sum\limits_{j=1}^{n} w_{ij}} \tag{1}$$

### 2.4. Cold and Hot Spots Analysis

Cold and hot spots analysis is a spatial clustering method that can show the distribution patterns of high and low-value spatial aggregation of indicators and compensate for the deficiencies of global spatial autocorrelation and natural break point grading for spatial characterization [29] by calculating the Getis-Ord $G_i^*$ statistic (called G-i-asterisk) for each

element in the dataset. In this paper, the $G_i^*$ coefficient proposed by Getis and Ord [54] is used to identify wetland cold and hot spots within the study area to explore where clustering of low or high wetland values occurs spatially, with the following expressions:

$$G_i^* = \frac{\sum_j^n w_{ij} x_j}{\sum_j^n x_j} \tag{2}$$

The $G_i^*$ test gives the following expression for $Z(G_i^*)$ [55].

$$Z(G_i^*) = \frac{\sum_j w_{ij} x_i - \overline{x} \sum_j^n w_{ij}}{S \sqrt{\frac{n \sum_j^n w_{ij}^2 - \left(\sum_j^n w_{ij}\right)^2}{n-1}}} \tag{3}$$

$$\overline{x} = \frac{\sum_j^n x_j}{n}, S = \sqrt{\frac{\sum_j^n x_j^2}{n} - (\overline{x})^2} \tag{4}$$

In the four expressions above, the spatial weight matrix is the attribute value, the average of all attribute values, and $n$ is the number of spatial cells. The $G_i^*$ statistic returned for each element in the dataset is the z-score. For a statistically significant positive z-score, the higher the z-score, the tighter the clustering of higher values (hot spots). For a statistically significant negative z-score, the lower the z-score, the tighter the clustering of lower values (cold spots).

*2.5. Construction of Ecological Corridors*

2.5.1. The Possible Connectivity Index (*PC*)

The probable connectivity index can indicate the spreading probability of ecological patches within the range. The formula is given below:

$$I_{PC} = \frac{\sum\limits_{i=0}^{n} \sum\limits_{j=0}^{n} a_i \cdot a_j \cdot P_{ij}^*}{A_L^2} \tag{5}$$

where $n$ denotes the total number of patches in the landscape. $a_i$ and $a_j$ denote the area of patch $i$ and patch $j$, respectively, and $A_L$ is the total area of the study area, and $P_{ij}^*$ is the probability of direct dispersal of species in patch $i$ and patch $j$. $0 < I_{PC} < 1$.

2.5.2. The Overall Connectivity Index (*IIC*)

The overall connectivity index indicates the overall connectivity between patches in the landscape. The formula is as follows:

$$IIC = \frac{\sum_{i-1}^{n} \sum_{j-1}^{n} \left[(a_i a_j) / (1 = nl_{ij})\right]}{A_L^2} \tag{6}$$

where $n$ denotes the total number of patches in the landscape. $a_i$ and $a_j$ denote the area of patch $i$ and patch $j$, respectively, and $nl_{ij}$ denotes the number of connections between patch $i$ and patch $j$, and $A_L$ is the total area of the study area, $0 \leq IIC \leq 1$, $IIC = 0$, there are no connections between habitat patches; $IIC = 1$, the whole landscape is habitat patches.

2.5.3. The Plaque Importance Index

The patch importance index indicates the importance of individual patches in the whole, by which important patches can be identified. The formula is as follows:

$$dI = \frac{I - I_{remove}}{I} \times 100\% \tag{7}$$

where d*I* is the patch importance index; *I* is the possible connectivity index; $I_{remove}$ is the overall connectivity of the landscape after the removal of the patch.

### 2.5.4. The *MCR* Model

The *MCR* model was initially proposed by the Dutch scientist Knappen and subsequently applied to the study of biological diffusion processes. It is based on a given minimum cumulative resistance surface, which allows for the determination of the minimum consumptive pathway for a species during migration [33]. The formula is as follows:

$$MCR = f_{\min} \sum_{j=n}^{i=m} \left( D_{ij} \times R_i \right) \tag{8}$$

where $f$ is a proportional function between *MCR* and the variables $D_{ij}$ and $R_i$; $D_{ij}$ is the spatial distance of matter or energy from $j$ to $i$; and $R_i$ is the resistance value of landscape $i$.

In order to employ the *MCR* model, it is essential to generate the minimum cumulative resistance data. This study considers pertinent literature and the circumstances of the study area and selects five types of resistance factors: land-use status, NDVI, elevation, slope, and distance from the water bodies. The NDVI data were obtained from the Earth Resources Data Cloud Platform, and the elevation data were obtained from the Geospatial Data Cloud. The data utilized in the analysis of slope are derived from the processing of elevation data. The data on water bodies in distance from the water bodies were obtained from the National Catalogue Service For Geographic Information website. The data were resampled to a uniform 30 m resolution. A hierarchical analysis was employed to calculate the weights, combining expert scoring with the literature [34,56,57]. One resistance factor corresponds to a range of resistance values that goes from 1 to 9. Table 2 shows the detailed regulations and weights.

**Table 2.** Corresponding resistance values and weights of resistance factors.

| Drag Factor | Factor | Resistance Value | Weights |
|---|---|---|---|
| Land-use type | Woodland | 1 | |
| | River, reach, reed swamp | 3 | |
| | Farmed lake, paddy field | 5 | 0.4921 |
| | Dry land | 7 | |
| | Residential land | 9 | |
| NDVI | 0–0.2 | 9 | |
| | 0.2–0.4 | 7 | |
| | 0.4–0.6 | 5 | 0.2350 |
| | 0.6–0.8 | 3 | |
| | 0.8–1 | 1 | |
| DEM/m | 0–20 | 1 | |
| | 20–40 | 3 | |
| | 40–60 | 5 | 0.1222 |
| | 60–80 | 7 | |
| | 80–100 | 9 | |
| Elevation/° | 0–5 | 1 | |
| | 5–10 | 3 | |
| | 10–20 | 5 | 0.0993 |
| | 20–30 | 7 | |
| | 30–50 | 9 | |
| Distance from the water body/m | 0–200 | 1 | |
| | 200–500 | 3 | |
| | 500–1000 | 5 | 0.0514 |
| | 1000–2000 | 7 | |
| | >2000 | 9 | |

*2.6. Research Flowchart*

Figure 2 depicts the research process and the principal findings of this study. Firstly, the distribution of wetlands and the HPI within the regional grid was determined. Secondly, the correlation between the two variables was analyzed in order to identify the most significant wetland resources within the study area. Finally, a recommendation for optimizing the landscape pattern in the study area was proposed based on the findings.

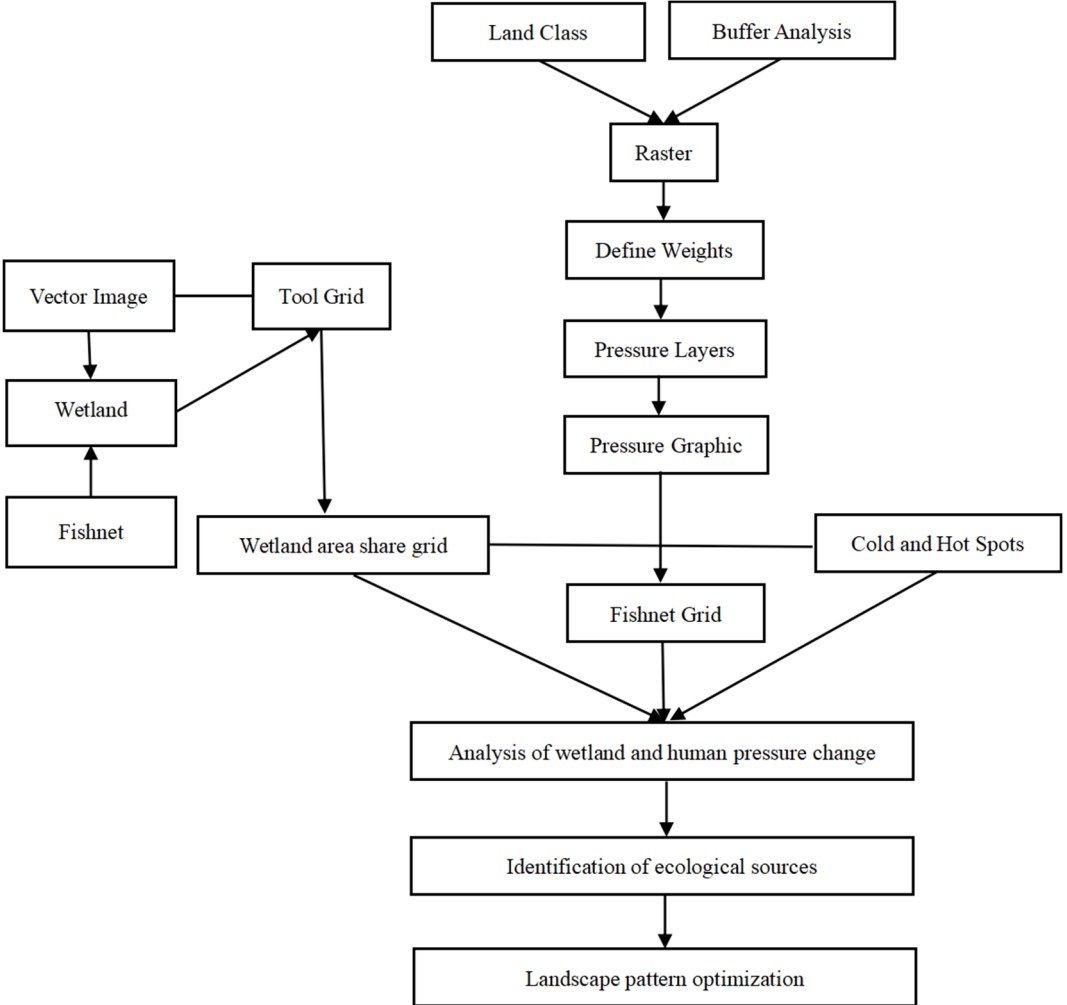

**Figure 2.** Research flowchart.

## 3. Results

### 3.1. Analysis of the Spatial Distribution of the Linghekou Wetland Resource

In order to gain a more intuitive understanding of the spatial distribution pattern of wetland resources and the changes in wetland resources in the Linghekou wetland, with a view to formulating effective policies for the sustainable development and conservation of wetland resources, this study identifies the wetland resources in the study area based on the land-use data of the Linghekou wetland in four periods: 2006, 2010, 2016, and 2020. The spatial distribution patterns of wetland resources for the aforementioned four periods are illustrated in Figure 3. The findings of the study indicate that the shrinkage of wetland resources in the Linghekou wetland was considerable between 2006 and 2020. This was evidenced by the transformation of numerous scattered, severely degraded wetlands into fully degraded wetlands, accompanied by a notable reduction in the area of wetlands without degradation. The most pronounced degradation and shrinkage of wetlands were observed in the central, northern, and southwestern regions of the study area.

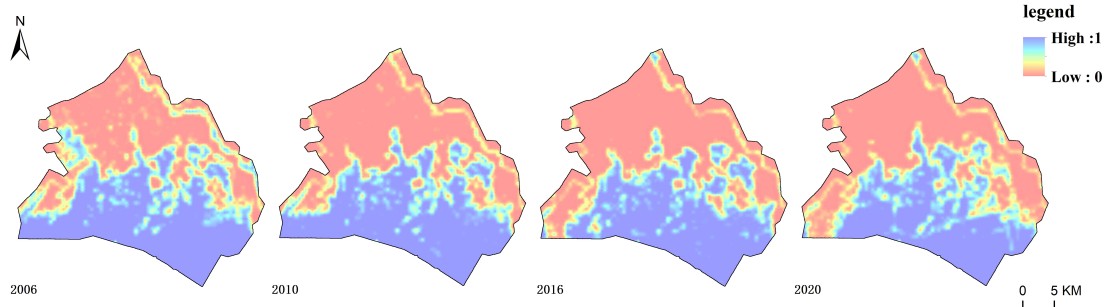

**Figure 3.** Spatial distribution of the Linghekou wetland resources in 2006, 2010, 2016, and 2020.

As demonstrated in Table 3, further quantitative analysis of the study results indicates that the extent of completely degraded wetlands in the Linghekou wetland increased between 2006 and 2020. Specifically, the percentage of wetlands within Linghekou wetland that are set to disappear between 2006 and 2020 is estimated to be 13.87%, rising from 23.11% in 2006 to 36.98% in 2020. This represents a notable reduction in wetland resources. Concurrently, the percentages of degraded, slightly degraded, potentially degraded, and non-degraded areas are expected to decline. This decline is expected to be accompanied by a reduction in the proportion of severely degraded areas within the total wetland network, which is projected to decline from 18% in 2006 to 12.48% in 2020. The proportion of the wetland network that is degraded is projected to increase from 15% in 2006 to 12.48% in 2020. Concurrently, the proportion of the wetland network that remains undegraded will decrease from 40.26% in 2006 to 33.30% in 2020.

**Table 3.** Grid proportion of the Linghekou wetland resources in 2006, 2010, 2016, and 2020.

| Wetland Percentage | Classification | 2006 | 2010 | 2016 | 2020 |
|---|---|---|---|---|---|
| 0 | Completely degradation | 23.11% | 34.62% | 39.29% | 36.98% |
| 0–20% | Severe degradation | 18.15% | 11.14% | 9.29% | 12.48% |
| 20–40% | Degradation | 6.67% | 5.44% | 5.90% | 6.44% |
| 40–60% | Slightly degradation | 5.73% | 4.44% | 4.19% | 5.01% |
| 60–80% | Potential degradation | 6.10% | 4.70% | 5.01% | 5.78% |
| 80–100% | Non-degradation | 40.26% | 39.66% | 36.32% | 33.30% |

The results of the data analysis indicate that a significant quantity of wetland resources has been lost in the Linghekou wetland. Furthermore, the health and stability of the wetland ecosystem have been significantly compromised by human activities and climate change during the study period. However, the Linghekou wetland ecosystem did not continue to deteriorate between 2006 and 2020. Indeed, the area of completely degraded wetlands decreased from 39.26% to 36.98% between 2016 and 2020, and the overall area of the Linghekou wetland recovered. Consequently, the implementation of reasonable and effective eco-environmental protection measures and human activity restriction policies contributes to the sustainable development of the wetland ecosystem and the restoration of the natural environment.

### 3.2. Changes in the Spatial Distribution of the HPI in the Linghekou Wetland

In order to analyze the impact of human activity intensity on the spatial distribution pattern of wetland resources in different periods and to enhance the reliability and scientific validity of this study, this paper employs a variety of data sources, including land-use type data, road distribution data, railway distribution data, navigable water distribution data, population density data, and lighting data, to calculate the HPI in 2006, 2010, 2016, and 2020 for the Linghekou wetland. The HPI of the Linghekou wetland was calculated for four periods in order to analyze the trends of human activity intensity and spatial distribution characteristics. As illustrated in Figure 4, the results of the study indicated

that the HPI of the Linghekou wetland in 2006, 2010, 2016, and 2020 were 20.31, 20.34, 21.22, and 22.29, respectively. The HPI of the Linghekou wetlands as a whole exhibited an increasing trend between 2006 and 2020. The most pronounced increase in human activity intensity was observed in the northwestern region of the Linghekou wetland, while the HPI decreased in the central, southern, and eastern regions of the study area. Consequently, the establishment of the Linghekou wetland nature reserve has only led to an improvement in the ecological environment in some areas and has not sufficiently curbed the impact of human activities on the area.

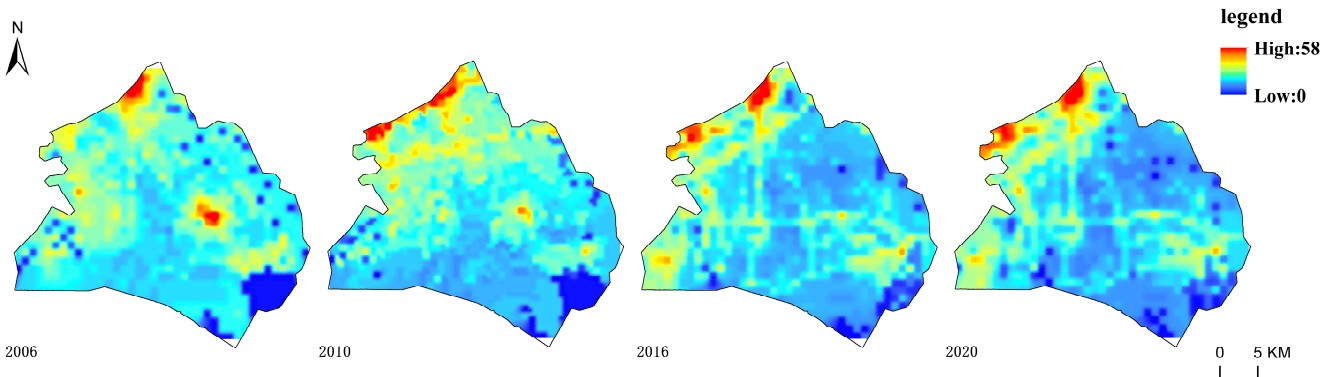

**Figure 4.** Distribution of the HPI in the Linghekou wetland in 2006, 2010, 2016, and 2020.

Further analysis of the HPI results for the Linghekou wetland in 2006, 2010, 2016, and 2020 indicates that the intensity of human activities within the Linghekou wetland remained relatively consistent between 2006 and 2010. However, by 2016, the intensity of human activities in the northwestern and southwestern regions of the study area had increased significantly. The HPI also exhibited an increase in the central, southern, and eastern regions of the Linghekou wetland in 2020 in comparison to the other three periods. This was accompanied by a decrease in the impact of human activities on the wetland. However, the HPI exhibited an increase in other areas.

*3.3. Analysis of Changes in the Spatial Distribution of Cold and Hot Spots in the Linghekou Wetland*

In order to better analyze the changes in the spatial distribution pattern of the Linghekou wetland, this study calculated the distribution pattern of cold and hot spots in the Linghekou wetland for four periods: 2006, 2010, 2016, and 2020. This was performed using the cold and hot spots analysis function of ArcGIS, and the calculation results are shown in Figure 5 and Table 4. The analysis results indicate that the cold spots, sub-cool spots, and hot spots in the Linghekou wetland are undergoing a state of continuous transformation due to human activities in the aforementioned four periods. The overall transformation trend can be described as follows: hot spots are transformed into sub-cool spots, sub-cool spots are transformed into cold spots, and the area covered by cold spots is continuously expanded. Conversely, the proportion of cold spots decreased from 13.39% to 10.46%, while the area covered by hot spots decreased by 1.54%. This phenomenon indicates that between 2006 and 2020, the Linghekou wetland was under increasing pressure from human activities, and the wetland ecosystem faced significant challenges in terms of environmental protection.

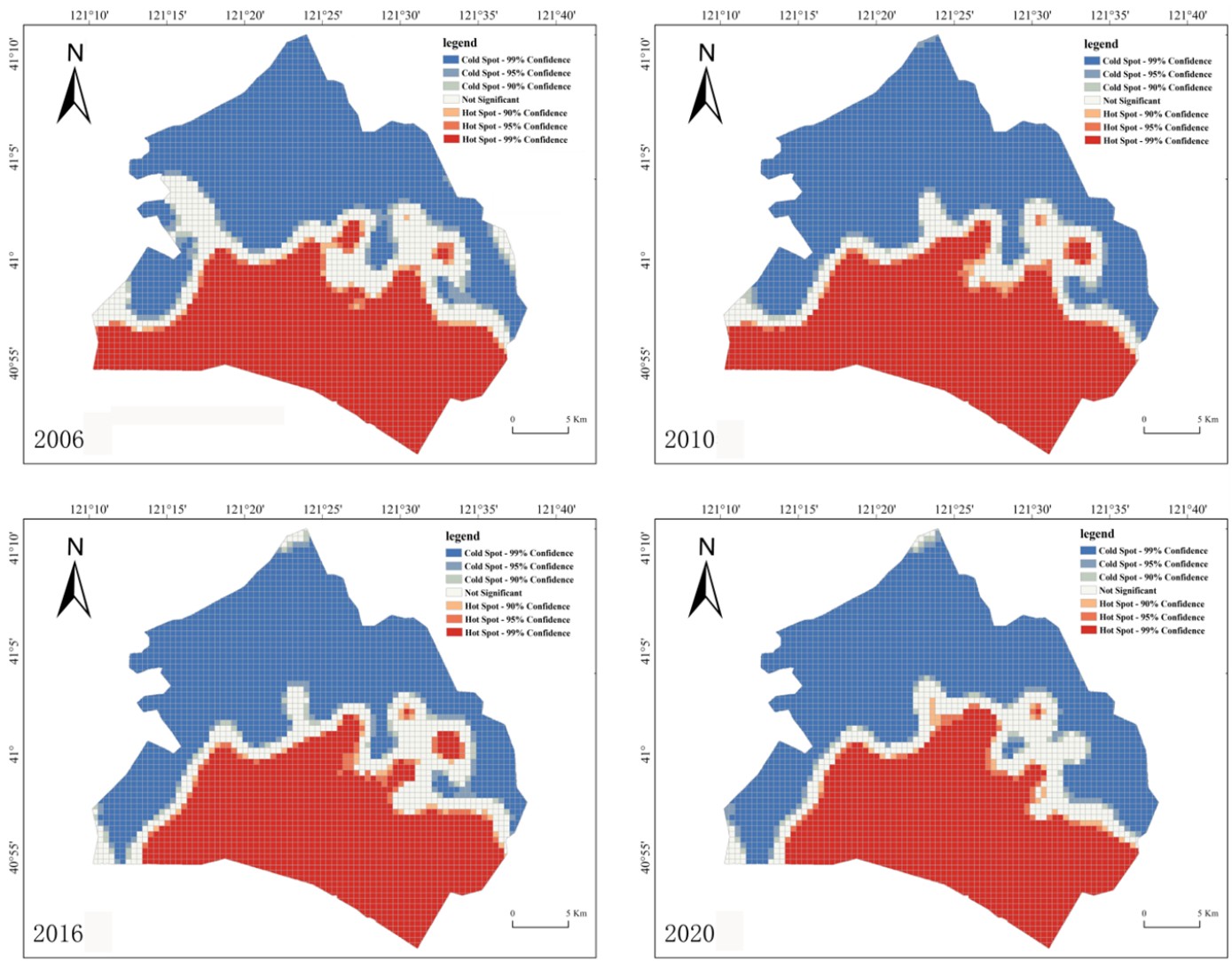

**Figure 5.** Spatial distribution of the Linghekou wetland cold and hot spots in 2006, 2010, 2016, and 2020.

**Table 4.** Proportion of cold and hot spots area of the Linghekou wetland in 2006, 2010, 2016, and 2020.

| Cold and Hot Confidence Level | Cold or Hot Spots | 2006 | 2010 | 2016 | 2020 |
|---|---|---|---|---|---|
| Confidence level 99% cold spot | Cold spots | 41.37% | 44.56% | 45.50% | 46.75% |
| Confidence level 95% cold spot | Cold spots | 2.91% | 2.11% | 2.22% | 2.22% |
| Confidence 90% cold spot | Cold spots | 1.45% | 1.17% | 1.34% | 1.23% |
| Non-significant | Sub-cold spots | 13.39% | 9.83% | 11.08% | 10.46% |
| 90% confidence in hot spots | Hot spots | 1.17% | 0.97% | 0.94% | 1.03% |
| 95% confidence in hot spots | Hot spots | 1.54% | 1.74% | 1.74% | 1.40% |
| 99% confidence in hot spots | Hot spots | 38.18% | 39.63% | 37.18% | 36.92% |

Further analysis of the statistical results of the cold and hot spot areas in different periods reveals that between 2006 and 2010, the cold spot area in the Linghekou wetland exhibited the fastest growth rate, increasing from 45.73% to 47.84%. Concurrently, the hot spot area also increased from 40.89% to 42.34% during this period. In conjunction with the findings presented in Figure 5, it becomes evident that during this period, the sub-cold spot within the study area underwent a transformation, evolving into both. Between

2010 and 2016, the cold spots in the Linghekou wetland extended to the southwest, and some of the hot spots in the southwest were transformed into cold spots or sub-cool spots as a result of the combined influence of human activities and climate change. The rate of degradation was the most rapid, while the percentage of hot spot area in the Linghekou wetland decreased by only 0.51% between 2016 and 2020.

The results of the aforementioned analysis demonstrate that the wetland resources of the Linghekou wetland are still being degraded by the continuously increasing pressure of human activities. Furthermore, the stability of the wetland ecosystem is low, and the wetland cold and hot spot area is in an unstable, fluctuating state. In order to ensure the sustainable development of the wetland ecosystem of the Linghekou wetland and to increase the stability of the wetland environment, it is necessary to provide a scientific reference for the development and use of the wetland environment through measures such as the rational adjustment of land-use types and structures and the formulation of relevant wetland protection policies.

*3.4. Identification and Analysis of Important Wetlands in the Linghekou Wetland*

A comprehensive analysis of the spatial and temporal changes in wetland resources, based on the results of the HPI calculation and the spatial distribution of wetlands in the Linghekou wetland, reveals that between 2006 and 2020, the wetland resources in the northwestern part of the study area will undergo a complete transformation from an interspersed distribution of degraded wetlands to a fully degraded wetland pattern. The high intensity of human activity has had a significant impact on the distribution of wetland resources in the northwestern part of the Linghekou wetland. The analysis of the HPI and the distribution of wetland resources in different periods between 2006 and 2020 indicates that the HPI in the southwestern part of the Linghekou wetland has been increasing between 2006 and 2016. This has resulted in the undegraded wetlands in this area being transformed into severely degraded wetlands. However, between 2016 and 2020, as the HPI in the east-central region of the Linghekou wetland decreased over this period, the wetland cover within the grid in this region underwent a significant transformation. Some fully degraded wetlands underwent a transition to a degraded state, while the number of grids comprising degraded and slightly degraded wetlands increased.

In conjunction with Figure 6, wetlands with a significant land area, experiencing human pressure and in a state of aggregation, are considered critical wetlands. The study demonstrates a spatial agglomeration among wetland distributions, with hot spots situated in the southern part of the study area and exhibiting a higher proportion of wetlands. From 2006 to 2020, the southwestern region of the study area exhibited a notable transformation of wetland hot spots into regular wetlands, accompanied by a concurrent degradation of the wetlands. Although the wetlands have not been entirely degraded and possess some ecological functions, the area is subject to significant human pressure, with an index range of 40 to 60. It was found that there has been a notable decline in pressure in the northern section of the region, with degraded wetlands corresponding to an area of increased human pressure. In consideration of the alteration of the cold and hot spots of the Linghekou wetland as a whole and the natural wetlands between 2006 and 2020 (illustrated in Figure 7), natural wetlands exceeding an area of 0.3 km$^2$ were identified as significant wetlands. Ultimately, 11 natural wetlands were identified as significant, and all types of change were maintained as hotspot areas between 2006 and 2020.

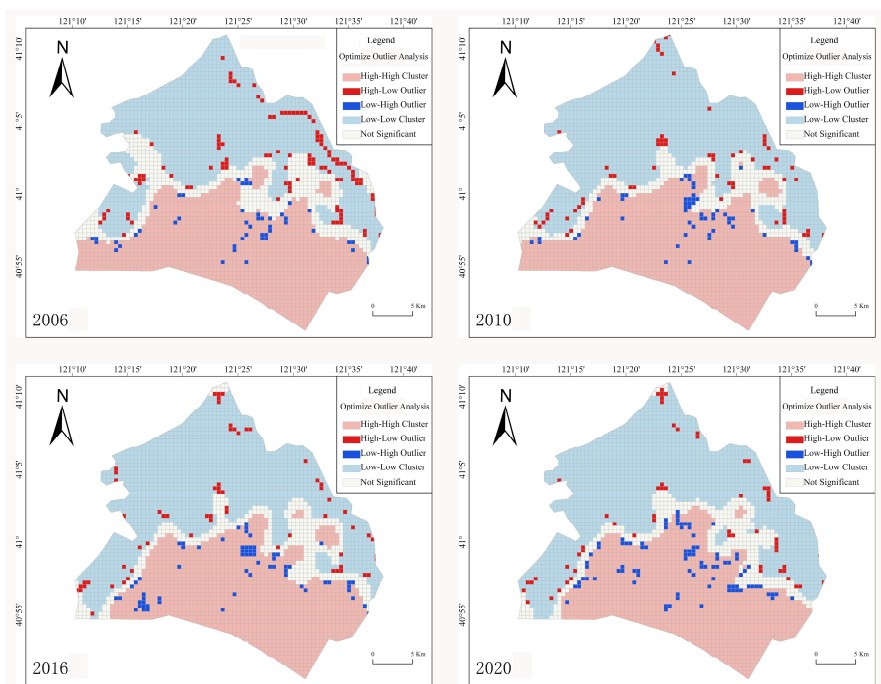

**Figure 6.** Spatial distribution of optimized anomalies of the Linghekou wetland in 2006, 2010, 2016, and 2020.

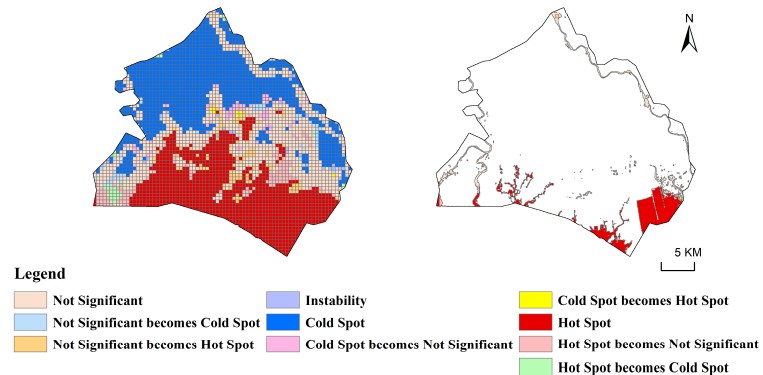

**Figure 7.** Changes in cold and hot spots in overall and natural wetlands in the Linghekou wetland.

### 3.5. Extracting the Ecological Source of the Linghekou Wetland

The test for connectivity of 11 vital wetlands, as shown in Table 5, involved adjusting the distance threshold in a constant manner, observing changes in PC and IIC values, and ultimately establishing a threshold of 2 km. Subsequently, as illustrated in Table 6, the probability was adjusted continuously, the rule governing the change in PC was observed, and the probability of 0.5 was determined. Subsequently, the PC and IIC were calculated between patches, and a significant connectivity index was established. This index was computed by weighting the dPC and dIIC equally at 0.5. The significant connectivity index, which identified wetlands with values exceeding 0.5 as ecological sources, yielded a total of six patches. Figure 8 illustrates that the majority of the ecological source sites were concentrated in the southern part of the study area. This concentration was primarily due to the location of the southern part of the study area by the sea, which provides optimal conditions for species to thrive and reproduce. The total area of the ecological source sites was 36.47 km², which represented 4.35% of the study area. The area of greatest extent, measuring 24.66 km², is situated in the southeastern region of the study area. The predominant landscape type is reed swamp.

**Table 5.** Distance threshold determination.

| | Distance/m | | | | | | |
|---|---|---|---|---|---|---|---|
| | **100** | **200** | **500** | **1000** | **2000** | **3000** | **4000** |
| IIC | 0.44 | 0.45 | 0.45 | 0.45 | 0.59 | 0.6 | 0.6 |
| PC | 0.46 | 0.46 | 0.49 | 0.56 | 0.64 | 0.69 | 0.73 |

**Table 6.** Probability determination.

| | Probability | | | | | | | | |
|---|---|---|---|---|---|---|---|---|---|
| | **0.1** | **0.2** | **0.3** | **0.4** | **0.5** | **0.6** | **0.7** | **0.8** | **0.9** |
| PC | 0.5 | 0.54 | 0.57 | 0.6 | 0.64 | 0.68 | 0.73 | 0.8 | 0.9 |

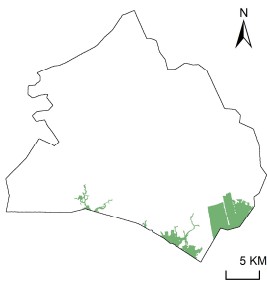

**Figure 8.** Distribution of ecological sources in the Linghekou wetland.

### 3.6. Wetland Ecological Corridor Construction in the Linghekou Wetland

The initial step is to construct the minimum cumulative resistance surface, which is achieved by combining land-use status, NDVI, elevation, slope, and distance from water bodies. The data must be reclassified using the ArcGIS software version 10.8.0.12790 in order to determine the resistance values of different value ranges (see Figure 9). Subsequently, hierarchical analysis is employed to ascertain the weights, which are then applied to the map algebra function of the ArcGIS software in order to determine the minimum cumulative resistance surface. As illustrated in Figure 10, the Linghekou wetland exhibits a maximum resistance value of 8.114, with built-up areas exhibiting larger resistance values.

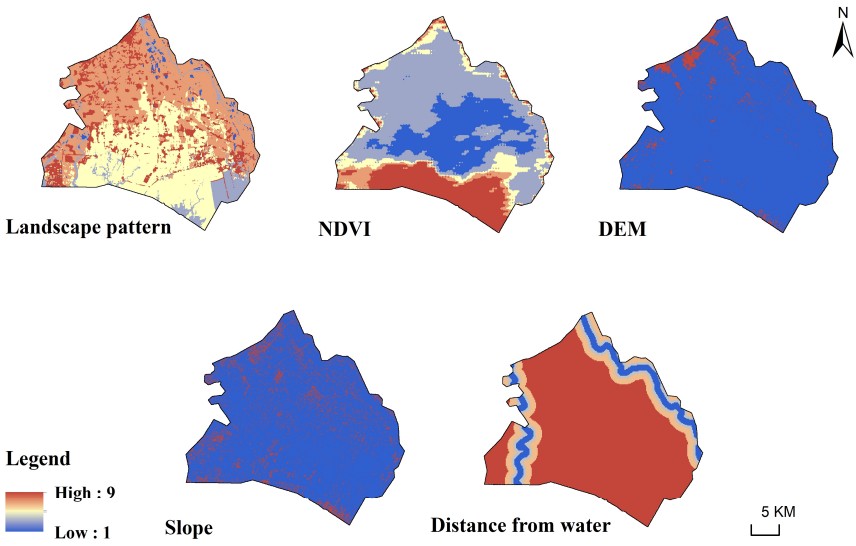

**Figure 9.** Resistance factor and resistance values.

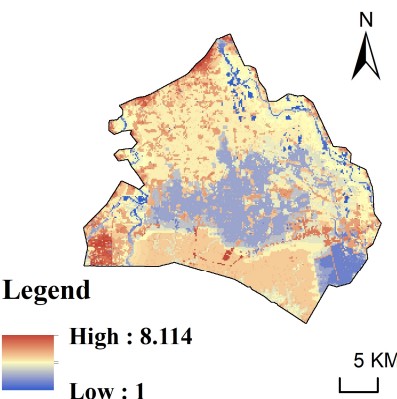

**Figure 10.** Minimum cumulative resistance surface.

The Linkage Mapper tool of ArcGIS software was employed to calculate the ecological corridors between the patches, utilizing the ecological source data and the minimum cumulative resistance surface data as the database. As illustrated in Figure 11, the analysis identified eight distinct ecological corridors, collectively spanning 50.93 km in length. The longest of these corridors is 20.81 km in length.

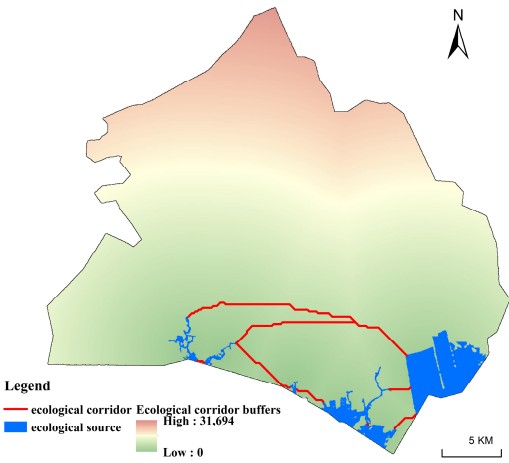

**Figure 11.** Wetland ecological corridor at the Linghekou wetland.

Human activities exert a profound influence on the spatial distribution of wetlands. An increase in human activities has resulted in a reduction of wetland area and proportion within the Linghekou wetland network, leading to a shrinkage of the wetland area and a decreased stability of the regional ecosystem. Conversely, a reduction in human activity intensity would facilitate the recovery of the wetland ecosystem. The establishment of ecological corridors can facilitate the protection of species migration routes. It is noteworthy that buffer zones may also emerge from the deployment of ecological corridors, wherein human activity is constrained. The ecological corridor in this study area traverses primarily paddy fields and farmed lakes, which are artificial wetlands that are intimately connected to human activities. To support this ecosystem, the government could implement policies such as leaving some grain in the fields during the autumn and purposefully leaving some small fish and shrimp in the ponds during salvage. Such measures will not only enhance the yield of crops in the forthcoming year but will also provide energy for birds during their migratory periods.

### 3.7. Identification of Ecological Nodes in the Linghekou Wetland

The ecological nodes identified in this paper are ecological pinch points and ecological barrier points. The main method is based on the ecological source data and the minimum

cumulative resistance surface data, applying the Linkage Mapper tool of the ArcGIS software to generate the corresponding current intensity, and extracting the high current areas as ecological nodes. This is illustrated in Figure 12. Among the ecological nodes of the Linghekou wetland, there are 32 ecological pinch points, and the main distribution of the landscape type is farmed lakes and paddy fields. The area in question comprises 10 ecological barrier points, with the predominant landscape type being farmed lakes.

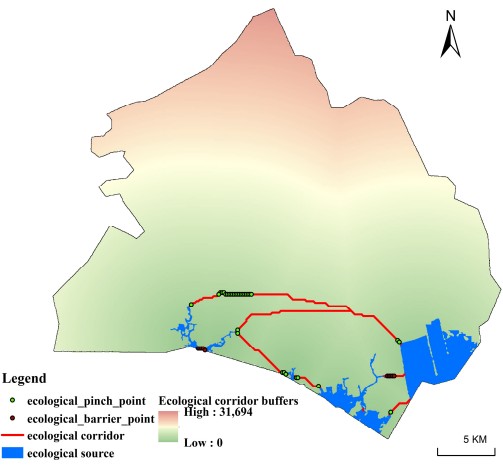

**Figure 12.** Wetland ecological corridor and ecological node at the Linghekou wetland.

### 3.8. Optimization of Landscape Pattern of the Linghekou Wetland

According to Linghai City's land planning for the next few years, applying the natural discontinuity point method, as illustrated in Figure 13. This method divides the minimum cumulative resistance surface into four classes, with the smallest resistance corresponding to the ecological core area, followed by the ecological buffer area, the ecological transition zone, and finally the living and working area. The ecological core area is distinguished by the lowest resistance value. Its predominant landscape types are natural wetlands, including rivers and reed swamps, which are highly sensitive to ecological change and provide valuable ecosystem services. Consequently, it is essential to enhance the protection of this area through artificial means, including the delineation of its boundaries and the regulation of human activities within it. It is recommended that the ecological buffer area be strengthened in the event that the protection of the ecological core area has spare capacity. This is due to the fact that the ecological buffer area has lower to moderate resistance values. The landscape pattern in the ecological buffer zone is paddy fields, which are an indispensable component of the agricultural lands in the study area. However, it is essential to strengthen the area's agroecological practices, with a particular focus on transforming the middle and low-yield fields to enhance the quality of arable land. Additionally, there is a need for a gradual conversion of irrational agricultural land to wetlands. The ecological transition zone serves as a transition zone between the ecological land and the production land. When human activities occur, the ecological transition zone plays a transitional role in the expansion of human activities. The area is home to a variety of ecological sources and corridors, with the majority of the landscape comprising dry land and farmed lakes. Given this, it is crucial to plan the land in a rational manner. In order to enhance the ecological value of the study area, it is recommended that some of the farmed lakes be reverted to their original state, that is, to the beach landscape. The living and working area has the highest resistance value, and the main landscape types are residential lands and dry lands. The distribution of wetland landscapes is very small because the area has the strongest human activities. Therefore, it is recommended that the construction of the living and working area be strengthened and that the suitability of human residential life in the area be improved.

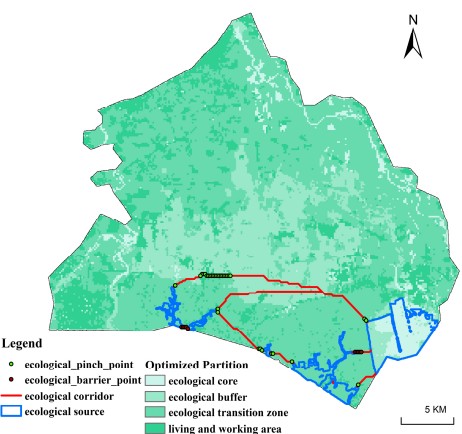

**Figure 13.** Landscape pattern optimization at the Linghekou wetland.

## 4. Discussion

The objective of this study is to investigate the relationship between human pressure and the distribution of wetland landscapes, employing the HPI method. This approach considers natural, economic, and social factors, integrating spatial autocorrelation and cold-hotspot analysis. The aim is to elucidate landscape evolution in the context of mounting human pressure and to provide a scientific foundation for the protection and utilization of wetland resources. Additionally, the study aims to advance our understanding of the impacts of human activities on the evolution of wetland landscapes in the Linghekou wetland. Our findings indicate that the spatial and temporal variability of HPI values in the Linghekou wetland is evident. From a spatial perspective, between 2006 and 2016, the western region exhibited relatively high HPI values due to the influence of the continuous expansion of residential land and anthropogenic interference, which increased year by year. However, from 2016 to 2020, the introduction of wetland protection policies by the local government and the implementation of protection measures in the Linghekou wetland reserve led to an improvement in the wetland environment and a significant reduction in the degradation degree. The results of this study demonstrate that changes in human pressures are an important driver of changes in wetland landscapes. Furthermore, the agreement with the results of previous studies provides the basis for the reliability of the research methodology employed in this paper [58,59].

In contrast to the previous optimization of landscape pattern by the *MCR* model, which directly designates ecological source sites or applies the morphological spatial analysis (MSPA) model [60,61], this study combines spatial autocorrelation and cold and hot spots analysis to select the patches in the natural wetlands that exhibit persistent hotspots and constant anthropogenic perturbation as ecological source sites. This approach improves upon the original ecological source site selection method and renders the results of the study more reasonable. Moreover, in contrast to the comprehensive optimization strategy proposed by previous studies [62,63], this study employs an ecological network framework based on the concept of "source-corridor-pinch point," integrating it with local land planning policies. This approach enables the formulation of tailored optimization strategies for distinct areas, resulting in a more comprehensive and coherent optimization of wetland landscapes. The findings of this research not only contribute to the existing body of scientific knowledge on the evolution of coastal wetlands under the influence of human activities but also provide insights into the scientific management and sustainable development of these ecosystems. However, due to the low resolution of the basic data used in this study, the accuracy of the HPI result values may be somewhat limited. Furthermore, a hierarchical analysis approach was employed in this study to determine the relative importance of the resistance factors. Some of the factors involved expert judgments, which inherently entail a certain degree of subjectivity. Consequently, future studies should investigate alternative weighting methodologies that offer greater accuracy.

## 5. Conclusions

This paper examines the spatial distribution relationship between the HPI and wetland resources in the Linghekou wetland, analyzes the significance of wetlands as ecological sources and applies the *MCR* model to optimize the landscape pattern of the Linghekou wetland. The principal findings of this study are as follows:

(1) 2006–2020, the area of non-degraded wetlands exhibited a decline, whereas other degraded wetlands with varying degrees of degradation demonstrated a rebound trend from the decrease. Concurrently, although the average value of the HPI has increased, this was concentrated in the southwestern part of the study area, which was also better controlled. Furthermore, the data indicates that the implementation of reasonable and effective ecological environmental protection measures and human activity restriction policies can facilitate the restoration of wetlands.

(2) The analysis of spatial autocorrelation and cold and hot spots for the period 2006–2020 indicates that the hot spots of wetland distribution are concentrated in the south of the study area. This reflects the abundance of wetland resources in the area, the high concentration of wetlands, and the low human pressure. In contrast, the cold spots of wetland distribution are primarily located in the northern part of the study area. This reflects the scarcity of wetland resources in the region, the weak concentration of wetlands, and the high human pressure. The combination of the two spatial relationships allows for the inference that the most significant wetlands in 2020 were situated in the southern region of the study area. Consequently, it becomes increasingly important to formulate policies related to human density or transportation restrictions, which are essential for the protection and restoration of wetlands in the study area.

(3) The study identified six ecological source sites in the Linghekou wetland, with a total area of 36.47 km$^2$, representing 4.35% of the study area. The largest of the identified sites is situated in the southeast corner of the study area, with an area of 24.66 km$^2$, and is classified as a reed swamp. Eight ecological corridors have been identified within the study area, with a total length of 50.93 km. The longest of these corridors is 20.81 km in length. The *MCR* model indicates that optimizing the zoning of the study area and proposing a landscape pattern based on land planning can effectively regulate human activities and protect wetland resources and biodiversity.

**Author Contributions:** Conceptualization, Q.C.; Methodology, M.W. and Q.C.; Software, M.W. and R.C.; Validation, M.W.; Formal analysis, Q.C.; Investigation, M.W.; Resources, Q.C.; Data curation, M.W. and R.C.; Writing—original draft, M.W. and R.C.; Writing—review & editing, Q.C.; Visualization, M.W. and R.C.; Supervision, Q.C. All authors have read and agreed to the published version of the manuscript.

**Funding:** This research received no external funding.

**Informed Consent Statement:** All authors have read, understood, and have complied as applicable with the statement on "Ethical responsibilities of Authors" as found in the Instructions for Authors and are aware that with minor exceptions, no changes can be made to authorship once the paper is submitted.

**Data Availability Statement:** All data analyzed in the research are presented in the paper, and all of them can be used to give appropriate references.

**Conflicts of Interest:** The authors declare no conflicts of interest.

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
