# Peer review of "Identification of Important Wetlands and Optimization of Landscape Patterns Based on Human Pressure Index: The Case of the Linghekou Wetland in China"

_sustainability, doi:10.3390/su16104082_

Round 1
Reviewer 1 Report
Comments and Suggestions for Authors
The article is relevant and the topic is worthy of investigation. The manuscript is well structured, the methodology is rigorous, and the conclusions are consistent with the results obtained. Even so, the need for a series of improvements is detected both in the introduction and review of the state of the art section, and in the scientific discussion section. If these improvements are incorporated correctly, the manuscript may be considered for publication in the journal. Below I detail the issues that must be addressed by the authors.
LITERATURE REVIEW
The review of the State of the art is limited from a methodological point of view to explaining the proposed approach based on the human pressure index with MCR models. In this aspect, a prior reflection is missing on why this analysis is focused, from that perspective for the case of the selected wetland, citing current alternative approaches. Authors should first cite other existing approaches for analyzing anthropogenic pressure in wetlands (there are several, see for example NDVI type indices through geostatistics in https://doi.org/10.3390/ijgi10090630 or SSP and RCP scenario simulations in https://doi.org/10.1016/j.ecolind.2022.108749) and then justify why the approach taken may more interesting for this study.
DISCUSSION
The scientific discussion section is very brief and lacks a higher level of scientific and academic erudition. In fact, a true section of scientific discussion is missing in the text, since what is there is above an simple explanation of results. Consequently, this section should be extended and improved. In this improved section of scientific discussion, the authors must analyze to what extent the results obtained corroborate, contradict, or improve the results of previous studies on the subject, citing those studies to enhance scientific and academic scholarship of the manuscript. The limitations of the proposed methodological framework should also be addressed here from a self-critical approach, explaining what issues could be improved in future lines of research. Lastly, the policy implications of the results obtained should also be made clear and the level of scientific and academic erudition of the text raised a little, pointing out relevant case studies in other parts of the world, where this methodological framework could be applied.
Author Response
The article is relevant and the topic is worthy of investigation. The manuscript is well structured, the methodology is rigorous, and the conclusions are consistent with the results obtained. Even so, the need for a series of improvements is detected both in the introduction and review of the state of the art section, and in the scientific discussion section. If these improvements are incorporated correctly, the manuscript may be considered for publication in the journal. Below I detail the issues that must be addressed by the authors.
1.LITERATURE REVIEW
The review of the State of the art is limited from a methodological point of view to explaining the proposed approach based on the human pressure index with MCR models. In this aspect, a prior reflection is missing on why this analysis is focused, from that perspective for the case of the selected wetland, citing current alternative approaches. Authors should first cite other existing approaches for analyzing anthropogenic pressure in wetlands (there are several, see for example NDVI type indices through geostatistics in https://doi.org/10.3390/ijgi10090630 or SSP and RCP scenario simulations in https://doi.org/10.1016/j.ecolind.2022.108749) and then justify why the approach taken may more interesting for this study.
Our response: Thank you for your suggestion, this section has been revised in line 28 to 127, and the references you mention have been added to references 14 and 16 of this article.
2.DISCUSSION
The scientific discussion section is very brief and lacks a higher level of scientific and academic erudition. In fact, a true section of scientific discussion is missing in the text, since what is there is above an simple explanation of results. Consequently, this section should be extended and improved. In this improved section of scientific discussion, the authors must analyze to what extent the results obtained corroborate, contradict, or improve the results of previous studies on the subject, citing those studies to enhance scientific and academic scholarship of the manuscript. The limitations of the proposed methodological framework should also be addressed here from a self-critical approach, explaining what issues could be improved in future lines of research. Lastly, the policy implications of the results obtained should also be made clear and the level of scientific and academic erudition of the text raised a little, pointing out relevant case studies in other parts of the world, where this methodological framework could be applied.
Our response: Thank you for your suggestion, the discussion have been modified as follows: The objective of this study is to investigate the relationship between human pressure and the distribution of wetland landscapes, employing the HPI method. This approach considers natural, economic, and social factors, integrating spatial autocorrelation and cold-hotspot analysis. The aim is to elucidate the landscape evolution in the context of mounting human pressure and to provide a scientific foundation for the protection and utilization of wetland resources. Additionally, the study aims to advance our understanding of the impacts of human activities on the evolution of wetland landscapes in the Linghekou wetland. Our findings indicate that the spatial and temporal variability of HPI values in the Linghekou wetland is evident. From a spatial perspective, between 2006 and 2016, the western region exhibited relatively high HPI values due to the influence of the continuous expansion of residential land and the anthropogenic interference, which increased year by year. However, from 2016 to 2020, with the introduction of The implementation of wetland protection policies by the local government and the implementation of protection measures in the Linghekou wetland reserve led to an improvement in the wetland environment and a significant reduction in the degradation degree. This evidence supports the conclusion that changes in human pressure are an important driver of changes in wetland landscapes, a conclusion that is consistent with the results of previous studies.
In contrast to the previous optimization of landscape pattern by the MCR model, which directly designates ecological source sites or applies the Morphological spatial analysis (MSPA) model, this study combines spatial autocorrelation and cold and hot spots analysis to select the patches in the natural wetlands that exhibit persistent hotspots and constant anthropogenic perturbation as ecological source sites. This approach improves upon the original ecological source site selection method and renders the results of the study more reasonable. Moreover, in contrast to the comprehensive optimization strategy proposed by previous studies, this study employs an ecological network framework based on the concept of "source-corridor-pinch point," integrating it with local land planning policies. This approach enables the formulation of tailored optimization strategies for distinct areas, resulting in a more comprehensive and coherent optimization of wetland landscapes. The findings of this research not only contribute to the existing body of scientific knowledge on the evolution of coastal wetlands under the influence of human activities, but also provide insights into the scientific management and sustainable development of these ecosystems. However, due to the low resolution of the basic data used in this study, the accuracy of the HPI result values may be somewhat limited. Furthermore, a hierarchical analysis approach was employed in this study to determine the relative importance of the resistance factors. Some of the factors involved expert judgments, which inherently entail a certain degree of subjectivity. Consequently, future studies should investigate alternative weighting methodologies that offer greater accuracy.
Reviewer 2 Report
Comments and Suggestions for Authors
Wang et al. selected Linghekou Wetland as the research area and intent to identify important wetlands and optimize landscape patterns through a combination of human stress index, various mathematical analysis methods, and MCR models. The research method is feasible and well organized, and the conclusions obtained are crucial for the practice of wetland restoration and protection. However, the current manuscript is slightly rough in writing and expression, and there is still a lot of room for improvement before publication.
Major comments:
The Introduction requires large modifications. (1) Too many redundant statements need to be appropriately simplified; (2) Some existing research results or conclusions lack literature support.
Both the discussion and conclusion require major revisions. (1) The discussion is limited and superficial, lacking discussion with existing research results and explanations for the findings of the study; (2) There are too many conclusions and mixed conclusions and discussions.
The citation format of references in the text is incorrect. The format of the ending reference is inconsistent with the requirements of the RS journal.
There are many issues with language expression, and it is recommended to seek native English speakers to further improve the manuscript.
Minor comments:
The abstract lacks innovative expression.
Line 11: The Human Pressure Index, abbreviated as HPI, because it appeared second times in Abstract.
Line 14. Change ‘conclusion’ to ‘results’
Line 17-18 Present the main research findings instead of repeating the methods.
Line 22. Delete ‘the’ before ‘Human Pressure Index’
Line 13. What’s the full name of MCR model? Provide the original references in the text.
For all the references published in Chinese, better to mark ‘in Chinese with English abstract’
Line 28. What is the ‘these factors’. Suggest delete the first sentence.
Lines 33-34. Loss of references.
Line 49. HPI has been abbreviated. This also appears in the form of the entire process.
Is the source of all data in Table 2.2 explained in the text? How to handle the inconsistent resolution of the data?
Line 99. Change ‘study’ to ‘Study’
Line 137. Error citation format.
There are too many errors or inappropriate expressions in the text. I will not list them one by one.
Comments on the Quality of English LanguageExtensive editing of English language required
Author Response
Wang et al. selected Linghekou Wetland as the research area and intent to identify important wetlands and optimize landscape patterns through a combination of human stress index, various mathematical analysis methods, and MCR models. The research method is feasible and well organized, and the conclusions obtained are crucial for the practice of wetland restoration and protection. However, the current manuscript is slightly rough in writing and expression, and there is still a lot of room for improvement before publication.
Major comments:
1.The Introduction requires large modifications. (1) Too many redundant statements need to be appropriately simplified; (2) Some existing research results or conclusions lack literature support.
Our response: Thank you for your suggestion, this section has been changed in lines 28 to 127. We have simplified the redundant expressions and added the reference
2.Both the discussion and conclusion require major revisions. (1) The discussion is limited and superficial, lacking discussion with existing research results and explanations for the findings of the study; (2) There are too many conclusions and mixed conclusions and discussions.
Our response: Thank you for your suggestion, the discussion have been modified as follows: The objective of this study is to investigate the relationship between human pressure and the distribution of wetland landscapes, employing the HPI method. This approach considers natural, economic, and social factors, integrating spatial autocorrelation and cold-hotspot analysis. The aim is to elucidate the landscape evolution in the context of mounting human pressure and to provide a scientific foundation for the protection and utilization of wetland resources. Additionally, the study aims to advance our understanding of the impacts of human activities on the evolution of wetland landscapes in the Linghekou wetland. Our findings indicate that the spatial and temporal variability of HPI values in the Linghekou wetland is evident. From a spatial perspective, between 2006 and 2016, the western region exhibited relatively high HPI values due to the influence of the continuous expansion of residential land and the anthropogenic interference, which increased year by year. However, from 2016 to 2020, with the introduction of The implementation of wetland protection policies by the local government and the implementation of protection measures in the Linghekou wetland reserve led to an improvement in the wetland environment and a significant reduction in the degradation degree. This evidence supports the conclusion that changes in human pressure are an important driver of changes in wetland landscapes, a conclusion that is consistent with the results of previous studies.
In contrast to the previous optimization of landscape pattern by the MCR model, which directly designates ecological source sites or applies the Morphological spatial analysis (MSPA) model, this study combines spatial autocorrelation and cold and hot spots analysis to select the patches in the natural wetlands that exhibit persistent hotspots and constant anthropogenic perturbation as ecological source sites. This approach improves upon the original ecological source site selection method and renders the results of the study more reasonable. Moreover, in contrast to the comprehensive optimization strategy proposed by previous studies, this study employs an ecological network framework based on the concept of "source-corridor-pinch point," integrating it with local land planning policies. This approach enables the formulation of tailored optimization strategies for distinct areas, resulting in a more comprehensive and coherent optimization of wetland landscapes. The findings of this research not only contribute to the existing body of scientific knowledge on the evolution of coastal wetlands under the influence of human activities, but also provide insights into the scientific management and sustainable development of these ecosystems. However, due to the low resolution of the basic data used in this study, the accuracy of the HPI result values may be somewhat limited. Furthermore, a hierarchical analysis approach was employed in this study to determine the relative importance of the resistance factors. Some of the factors involved expert judgments, which inherently entail a certain degree of subjectivity. Consequently, future studies should investigate alternative weighting methodologies that offer greater accuracy.
the conclusions have been modified as follows:This paper examines the spatial distribution relationship between the HPI and wetland resources in the Linghekou wetland and analyzes the significance of wetlands as ecological sources and applies the MCR model to optimize the landscape pattern of the Linghekou wetland. The principal findings of this study are as follows:
(1) 2006-2020, the area of non-degraded wetlands exhibited a decline, whereas other degraded wetlands with varying degrees of degradation demonstrated a rebound trend from the decrease. Concurrently, although the average value of the HPI has increased, this was concentrated in the southwestern part of the study area, which was also better controlled. Furthermore, the data indicates that the implementation of reasonable and effective ecological environmental protection measures and human activity restriction policies can facilitate the restoration of wetlands.
(2) The analysis of spatial autocorrelation and cold and hot spots for the period 2006-2020 indicates that the hot spots of wetland distribution are concentrated in the south of the study area. This reflects the abundance of wetland resources in the area, the high concentration of wetlands, and the low human pressure. In contrast, the cold spots of wetland distribution are primarily located in the northern part of the study area. This reflects the scarcity of wetland resources in the region, the weak concentration of wetlands, and the high human pressure. The combination of the two spatial relationships allows for the inference that the most significant wetlands in 2020 were situated in the southern region of the study area. Consequently, it becomes increasingly important to formulate policies related to human density or transportation restrictions, which are essential for the protection and restoration of wetlands in the study area.
(3) The study identified six ecological source sites in the Linghekou wetland, with a total area of 36.47 km², representing 4.35% of the study area. The largest of the identified sites is situated in the southeast corner of the study area, with an area of 24.66 km², and is classified as a reed swamp. Eight ecological corridors have been identified within the study area, with a total length of 50.93 km. The longest of these corridors is 20.81 km in length. The MCR model indicates that optimizing the zoning of the study area and proposing a landscape pattern based on land planning can effectively regulate human activities and protect wetland resources and biodiversity.
3.The citation format of references in the text is incorrect. The format of the ending reference is inconsistent with the requirements of the RS journal.
Our response: Thank you for your suggestion, errors in citation of references and formatting of references have been amended.
4.There are many issues with language expression, and it is recommended to seek native English speakers to further improve the manuscript.
Our response: Thank you for your suggestion, language expression have been amended.
Minor comments:
1.The abstract lacks innovative expression.
Our response: Thank you for your suggestion, the abstract have been modified as follows: The Linghekou wetland is a rich repository of ecological resources and serves as an important habitat for numerous rare and protected animals. However, due to a confluence of natural and anthropogenic factors, the ecological environment of the Linghekou wetland is facing a multitude of threats, including the reduction of wetland area, the degradation of wetland resources, and the instability of ecological structure. This paper employs an anthropogenic focus, utilizing the Human Pressure Index (HPI), spatial autocorrelation, and cold and hot spots methods to identify crucial wetlands. These identified wetlands are then utilized as ecological source sites to optimize the landscape pattern of the Linghekou wetland, employing the minimum cumulative resistance (MCR) model. The final results indicated the identification of 6 ecological sources, 8 ecological corridors, and 42 ecological nodes. These were primarily concentrated in the southern region of the study area and were distributed in a reasonable manner. The method of identifying ecological sources when optimizing the landscape pattern with the MCR model was enriched by this approach. Additionally, the paper offers recommendations for the optimization of the landscape pattern of the Linghekou wetland and establishes a foundation for the protection and restoration of other similar wetlands.
2.Line 11: The Human Pressure Index, abbreviated as HPI, because it appeared second times in Abstract.
Our response: Thank you for your suggestion, it has been amended in line 13.
3.Line 14. Change ‘conclusion’ to ‘results’
Our response: Thank you for your suggestion, it has been amended in line 16.
4.Line 17-18 Present the main research findings instead of repeating the methods.
Our response: Thank you for your suggestion, it has been amended in line 16 to 18.
5.Line 22. Delete ‘the’ before ‘Human Pressure Index’
Our response: Thank you for your suggestion, it has been amended in line 24.
6.Line 13. What’s the full name of MCR model? Provide the original references in the text.
Our response: Thank you for your suggestion, the full name of the MCR model has been changed at line 15 and the original reference is introduced in lines 254 to 257.
7.For all the references published in Chinese, better to mark ‘in Chinese with English abstract’
Our response: Thank you for your suggestion, we have marked "in Chinese with English abstract" after the references published in Chinese.
8.Line 28. What is the ‘these factors’. Suggest delete the first sentence.
Our response: Thank you for your suggestion, the first sentence has been deleted.
9.Lines 33-34. Loss of references.
Our response: Thank you for your suggestion, it has been amended in line 35.
10.Line 49. HPI has been abbreviated. This also appears in the form of the entire process.
Our response: Thank you for your suggestion, we've checked the full paper and made changes.
11.Is the source of all data in Table 2.2 explained in the text? How to handle the inconsistent resolution of the data?
Our response: Thank you for your suggestion, data sources and processing have been refined in line 265 to 272.
12.Line 99. Change ‘study’ to ‘Study’
Our response: Thank you for your suggestion, it has been amended in line 130.
13.Line 137. Error citation format.
Our response: Thank you for your suggestion, it has been amended in line 169.
14.There are too many errors or inappropriate expressions in the text. I will not list them one by one.
Our response: Thank you for your suggestion, we've checked the full paper and made changes.
15.Comments on the Quality of English Language. Extensive editing of English language required.
Our response: Thank you for your suggestion, language expression have been amended.
Round 2
Reviewer 2 Report
Comments and Suggestions for Authors
Although the author has made extensive revisions, there are still some issues that need to be addressed:
1. The expression of existing research lacks literature support. For example, Lines 47-103.
2. Text display error. Before uploading the manuscript, should review it in detail. For example Line 556.
Author Response
- The expression of existing research lacks literature support. For example, Lines 47-103.
Our response: Thank you for your suggestion, it has been amended in line 48, 61 and 104. References 12-15, 18, 19, and 37-39 are new to the paper.
- Text display error. Before uploading the manuscript, should review it in detail. For example Line 556.
Our response: Thank you for your suggestion, it has been amended in line 555 to 558.